# Status exploration and analysis of regional hospice and palliative care networks in Germany: A protocol for a mixed-methods study

Sven Schwabe[1]*, Christoph Buck[2,3], Franziska A. Herbst[1], Tanja Schleef[1], Stephanie Stiel[1], Nils Schneider[1]

1 Institute for General Practice and Palliative Care, Hannover Medical School, Hannover, Germany, 2 Branch Business and Information Systems Engineering of the Fraunhofer FIT, University of Applied Sciences Augsburg, Augsburg, Germany, 3 Center for Future Enterprise, School of Management, Faculty of Law & Business, Queensland University of Technology, Brisbane, Australia

* schwabe.sven@mh-hannover.de

## Abstract

### Background

Multi-professional cooperation between healthcare providers is a key quality criterion of hospice and palliative care. While hospice and palliative care networks can support cooperation on a local level, opportunities for wider cooperation through the establishment and development of regional hospice and palliative care networks in Germany have not yet been explored systematically.

### Aims

The HOPAN study aims at: (1) identifying regional hospice and palliative care networks in Germany, (2) analysing these networks using an adapted quality assessment tool, and (3) proposing setting-sensitive recommendations for network development and exploring the benefits of these recommendations.

### Methods

HOPAN is a prospective, observational, mixed-methods study comprising three work packages (WPs). In WP1, the stock of regional hospice and palliative care networks in Germany will be identified via database, literature, and internet research. In WP2a, focus groups will be conducted to adapt an existing maturity model for healthcare networks to regional hospice and palliative care networks. In WP2b, a questionnaire will be sent to each identified regional hospice and palliative care network to gain insight into their structure and status of development. In WP2c, group discussions will be conducted to develop setting-sensitive recommendations for these networks. Finally, in WP3, these recommendations will be sent to all participating hospice and palliative care networks, and the benefits of the recommendations will be evaluated via a questionnaire.

relevant data from this study will be made available upon study completion.

**Funding:** The study "HOPAN – Status exploration and analysis of regional hospice and palliative care networks using an adapted quality assessment tool" is funded (on the basis of peer review) by the Innovation Fund of the German Federal Joint Committee (G-BA) (Grant N˚ 01VSF22042). The grant was awarded to SvS. The funders had and will not have a role in study design, data collection and analysis, decision to publish, or preparation of the manuscript.

**Competing interests:** The authors have declared that no competing interests exist.

## Discussion

Empirically developed setting-sensitive recommendations should enable the systematic establishment and management of regional hospice and palliative care networks in Germany, considering the specific needs and potential of each network. The study findings are expected to improve the overall development of hospice and palliative care services.

## Trial registration

The study was prospectively registered in the German Clinical Trials Register (Deutsches Register Klinischer Studien) (Registration N˚ DRKS00030629; date of registration: 02 November 2022). The study is searchable under the International Clinical Trials Registry Platform Search Portal of the World Health Organization, under the German Clinical Trials Register number.

## Introduction

Until recently, some federal states in Germany promoted hospice and palliative care networks on a voluntary basis; however, most networks were financed by the donations and fees of network members [1–5]. To improve on this situation, in 2021, the German legislature passed a new law (Gesetz zur Weiterentwicklung der Gesundheitsversorgung/GVWG) providing for the promotion of regional hospice and palliative networks [6]. The aim of this legislation is to improve multi-professional and cross-sectoral cooperation in hospice and palliative care at a regional level, via network coordinators, funded by statutory health insurance [7]. Specifically, coordinators are tasked to: (a) support cooperation and coordination between network members, (b) support joint public relations, (c) initiate and organise further and advanced training programmes, (d) organise regular network meetings, and (e) cooperate with other regional counselling services [8]. The networks should include all providers of general and specialised palliative care in the inpatient and outpatient setting. In Germany, the focus of hospice care is on psychosocial and bereavement support. Hospice care takes place in inpatient hospices, hospice day clinics and outpatient hospice services, which should be integrated into the networks [9].

A comprehensive overview of regional hospice and palliative care networks in Germany is lacking. To date, only two smaller surveys at the level of individual federal states have been conducted [3, 5]. One study published regional recommendations for the implementation and coordination of hospice and palliative care networks [10], showing that the investigated networks differed considerably regarding their stages of development, organisational structures, fields of activity and working methods [4].

In hospice and palliative care, healthcare providers and volunteers must collaborate to improve the quality of life of patients and their families [11]. Cooperation between providers in care networks enables a seamless interplay of all healthcare professionals and volunteers, thereby improving the continuity of patient care [12, 13].

On a regional level, collaboration between healthcare providers in hospice and palliative care seems to positively impact individual casework [14]. According to the literature, regional networks may increase the number of personal meetings between healthcare providers and increase members' willingness to cooperate [15]. Hence, regional networks may improve inter-professional communication among healthcare providers, evoke confidence in action, reduce stress among employees, ensure the early integration of psychosocial and spiritual

support, and reduce the number of hospital admissions [16]. They may also promote joint initiatives for the further development of local hospice and palliative care services and improve the organisation of patient-centred care [14, 15].

Until recently, the establishment and development of regional hospice and palliative care networks in Germany was largely unsystematic. These networks were not integrated into the German healthcare system and structural standards and quality indices for network collaboration and coordination were lacking.

## Materials and methods

### Study aim

The project "HOPAN–Status exploration and analysis of regional hospice and palliative care networks using an adapted quality assessment tool" has the following three objectives:

1. to identify regional hospice and palliative care networks in Germany,

2. to analyse these networks using an adapted quality assessment tool, and

3. to propose setting-sensitive recommendations for further network development and to explore the benefits of these recommendations.

### Design

HOPAN is a prospective, observational, mixed-methods study organised into three working packages (WPs). The study protocol adheres to STROBE guidelines [17]. WP1 will involve the identification of regional hospice and palliative care networks in Germany via database, literature and internet research. The findings will provide an overview of the number and regional distribution of hospice and palliative networks in Germany.

In WP2a, an existing maturity model for healthcare networks, developed within the INDiGeR research project, will be adapted to accommodate the structures and contents of the identified regional hospice and palliative care networks [18]. The INDiGeR research aimed at identifying the most relevant dimensions of successful network management and developing general recommendations for the implementation and improvement of healthcare and service networks. The project identified four key dimensions of network management: (a) infrastructure, (b) governance, (c) promotion, and (d) moderation (see Fig 1).

The maturity model focuses on the common provision of services from a single network perspective. This methodological tool will be adapted for application to hospice and palliative networks.

In WP2a, a group of experts will be formed on the basis of the survey results (WP1) with the leaders (e.g. coordinators, moderators) of hospice and palliative care networks in Germany and other network experts. Expert group workshops will examine the extent to which the maturity model should be supplemented and modified to accommodate hospice and palliative care networks, and how the four relevant network dimensions (i.e. infrastructure, moderation, governance, promotion) should be specified and operationalised. This specified and adapted maturity model will form the basis for the model-based network analysis in WP2b.

In WP2b, the adopted maturity model for regional hospice and palliative care networks will be operationalised into questionnaire items. Subsequently, an online survey will be administered to all of the identified networks. The survey is expected to comprise three sections:

a. A general section on the networks' basic and structural data (e.g. year of founding, number of partners, participating care actors, number of meetings per year).

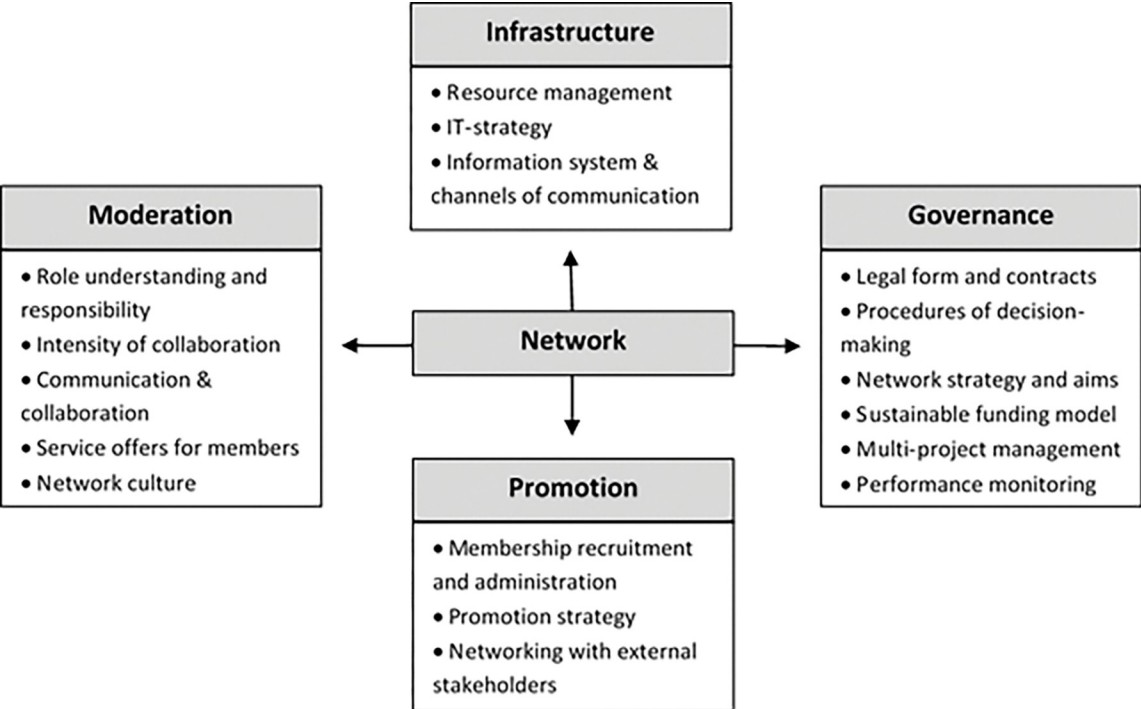

**Fig 1. Key dimensions of network management from the INDiGeR project [18] (authors' own translation into English).**

b. A specific section on the model-based network evaluation, focussing on network dimensions and the degrees identified in WP2a (e.g. for network moderation: role understanding and responsibility, intensity of cooperation, communication, services offered to network members, network culture). Specific items will be associated with each sub-dimension, graded in four levels (e.g. initial, structured, established, optimised). Based on the results, a developmental status will be assigned to each dimension.

c. A retrospective section on good practice and experiences of network development. Each dimension or sub-dimension will be specifically queried, to generate practical examples for the development of recommendations in WP2c.

In WP2c, four expert workshops with the abovementioned network leaders and coordinators ($n = 10$) will be conducted to develop setting-sensitive recommendations for regional hospice and palliative care networks, on the basis of the results from WP2b.

In WP3, the recommendations arising out of WP2c will be disseminated to individual networks and usability will be evaluated. Each network surveyed in WP2b will receive setting-sensitive, individual recommendations, according to the network's maturity level. Additionally, each network will receive a link to a quantitative, standardised, online questionnaire to evaluate the utility of the recommendations. This questionnaire will be developed a priori by the research team and will contain questions on the usefulness of the recommendations in their respective dimensions, as well as the actual and planned implementation of the recommendations. The link to the online questionnaire will be sent to the networks using the survey software SoSciSurvey (SoSci Survey GmbH, 2006–2022). The primary aim will be to explore the relevance and feasibility of the setting-sensitive recommendations. Finally, the maturity model and its level-specific recommendations will be adapted in accordance with the survey results and prepared for regional hospice and palliative care networks, so it can be used by: (a) further

regional hospices and networks for the purpose of self-assessment and (b) health insurance providers for the purposes of quality assessment and reporting.

## Study population and data collection

In WP1, all regional hospice and palliative care networks in Germany will be identified as completely as possible. The total stock of such networks in Germany is currently unknown. Hence, the size of this population can only be estimated. Based on the identified hospice and palliative care networks in Lower Saxony [19] and Bavaria [20], a total of 150 networks in Germany is estimated. Data will be collected via internet and literature research, supplemented by a search of national databases, such as the publicly accessible palliative care portal [21], the "Wegweiser Hospiz- und Palliativversorgung Deutschland" [22], and partly available lists of health ministries and state coordination offices. Network names, contact details, regions, and other relevant and available information (e.g. year of founding, legal formation, sponsorship) will be recorded.

The group of experts, which will be set up in WP2a, will include leaders ($n = 10$; e.g. coordinators, moderators) of hospice and palliative care networks in Germany and other network experts from hospice and palliative care facilities in Germany. Participating network leaders ($n = 7–8$) will be selected from the group of networks identified in WP1. The selection will follow the maximum variation sampling strategy, based on criteria that have not yet been defined (e.g. size, duration of existence, degree of professionalisation). Moreover, a small number of experts ($n = 2–3$) will be recruited from the group of network members, based on the suggestions of cooperating German professional hospice and palliative care associations. If experts decline to participate in the study, the expert group will be supplemented on the basis of the same sampling criteria.

A total of $n = 5$ expert workshops are planned, each involving $n = 10$ of the abovementioned leaders and coordinators. Four of these meetings will be conducted as online video conferences. The first workshop will explore the extent to which the structure of the INDiGeR maturity model should be augmented and modified to accommodate hospice and palliative care networks. Members of the research team will send literature on the maturity model to participants prior to the workshop, and then present the model to participants at the workshop. The subsequent three workshops will aim at concretising and operationalising the assumed relevant network dimensions (i.e. infrastructure, moderation, governance, promotion) for regional hospice and palliative care networks. The documented results of the workshops will be compiled, reflected upon, discussed, and integrated by the research team and further developed into a maturity model for regional hospice and palliative care networks. The developed model will then be presented to the experts for their feedback in a further online workshop. In the expert workshops, established methods for moderating group discussions (e.g. card queries, World Cafe) will be used. Each workshop will be 3 hours in duration, and will be audio recorded and transcribed verbatim.

In WP2b, the respective leaders/coordinators of an estimated $n = 150$ hospice and palliative care networks will be sent an online survey. A response rate of 70% is expected, and thus $n = 105$ valid data sets to include in the analysis. Notwithstanding this calculated response rate, the aim will be to include as many as possible of the networks identified in WP1, in order to increase statistical power, especially for the more detailed network analysis.

In WP2c, expert workshops ($n = 4$) with the abovementioned network leaders/coordinators ($n = 10$) will be conducted to develop setting-sensitive recommendations for regional hospice and palliative care networks. For this purpose, the results of the network analysis for the individual dimensions of the maturity model (including the level-specific locations of the

networks) will be presented as impulse speech. In group discussions, participants will develop specific recommendations for the further development of networks for each development level and sub-dimension. Each workshop will be 3 hours in duration, and will be audio recorded and transcribed verbatim. An overlap of participants in WP2a and WP2c will promote participants' long-term commitment to and identification with the project.

In WP3, all networks that participated in WP2b will be administered a quantitative online survey. A total of $n$ = 105 participating networks in WP2b and a response rate of 70% are assumed. Hence, 74 valid data sets are expected for inclusion in the analysis. Respondents will indicate on a 4-point verbal rating scale the extent to which they deem the individual recommendations relevant and feasible for their own network. They will also be afforded the opportunity to make suggestions for improvement, via free text comments.

## Inclusion and exclusion criteria

WP2a and WP2c will include leaders (e.g., coordinators, moderators) of hospice and palliative care networks in Germany and other network experts from the field of hospice and palliative care in Germany. All participants will be aged $\geq$18 years and have theoretical and/or practical knowledge and/or experience of hospice and palliative care networks in Germany.

After receiving detailed information about the type, content, and purpose of the study and their participation, experts will provide written informed consent to participate. Experts of all genders and ethnic backgrounds will be invited to participate.

Experts will be excluded from the study if they are insufficiently proficient in the German language to join group discussions and/or if they do not consent to participate.

## Ethical considerations

The study was approved on 20 August 2022 by the Ethics Committee of Hannover Medical School (N˚ 10424_BO_S_2022) and the appointed data protection officer of Hannover Medical School. Prior to administering the questionnaire, the researchers will provide eligible participants with detailed information about the study type, content, purpose, and duration.

All study participants will be informed in detail, orally and in writing, about the project aims and expected outputs, before confirming their participation in the project. Participation will only be possible when the individual explicitly agrees to participate in the study and signs a written consent form. Each participant will have the right to refuse or discontinue participation at any time without providing any reasons for doing so.

The study results (e.g. transcripts) will be stored pseudonymously on the secure servers of the MHH, in order to ensure personal data protection and prevent the results from being linked to individual participants.

## Data analysis

The data analysis will aim at describing the current stock of regional hospice and palliative care networks in Germany and determining setting-sensitive recommendations for the further development of these structures.

In WP1, data from the national database, internet, and literature searches will be transferred to a Microsoft Excel 2016 spreadsheet. Key characteristics of the identified institutions will be documented in a template, according to availability (e.g. network name, district, federal state, contact person, contact address, legal formation, year of founding, region [urban/rural], and number of members). Additionally, quantitative data will be analysed descriptively using IBM SPSS Statistics 27 (SPSS Inc., Chicago, IL, USA) for Windows. The survey will provide an

overview of the number and regional distribution of hospice and palliative care networks in Germany.

In WP2a, the expert workshops ($n = 5$) with $n = 10$ participants (each) will be audio recorded and transcribed verbatim. Visual data, including the findings of card queries and World Cafes, will be photographed. Both oral and visual data will be analysed according to qualitative content analysis, using MAXQDA 2022 (VERBI Software Consult Sozialforschung GmbH, 1989–2022) [23].

In WP2b, the results of the online survey will be analysed descriptively and with frequency tables, using IBM SPSS Statistics 27. If possible, cluster analyses will be conducted to identify similar structures and different network types (e.g. "public/outward-oriented networks" vs. "member/inward-oriented networks"). Free text answers regarding network management will be inductively categorised, summarised, and evaluated using frequency counting.

In WP2c, the transcripts of the expert group discussions will be analysed according to qualitative content analysis, using MAXQDA 2022. A deductive coding procedure will be used, whereby the levels and sub-dimensions of the maturity model specified in WP2a will function as categories. The research team will review the recommendations in terms of coherence and adapt the wording as needed.

In WP3, quantitative data from the online survey will be analysed descriptively, using IBM SPSS Statistics 27. Free text comments will be analysed descriptively, in terms of content.

Fig 2 provides an overview of the mixed-methods study design across the three WPs. The figure illustrates the methods that will be used to develop setting-sensitive recommendations for regional hospice and palliative care networks in Germany.

## Expected results

The main expected results are: (1) an overview of existing regional hospice and palliative care networks in Germany, (2) a maturity model for regional hospice and palliative care networks, (3) a model-based network analysis, and (4) setting-sensitive recommendations for the further development of each network. These findings will contribute to the further development of hospice and palliative care services in Germany, on a wider scale.

## Study risks

Access to experts and leaders/coordinators of hospice and palliative care networks can be difficult, depending on their workloads and basic attitudes towards and experience with research projects. Hence, reaching the required number of participants may be a methodological challenge. However, achievement of the case numbers seems feasible, since the majority of the WPs involving leaders/coordinators are based on a qualitative research design, and the case numbers were calculated generously. Also, recruitment will be broad.

For the nationwide identification of regional hospice and palliative care networks, cooperative agreements will be made with all of the relevant associations, ensuring access to the relevant databases [22] and network coordinators. In addition, the status analysis will be supplemented by an independent internet search for each federal state. This staggered procedure will ensure that networks are comprehensively identified and addressed. Furthermore, it is expected that the networks will have an intrinsic interest in participating in the project, in order to benefit from the specific recommendations for network development.

## Study status and timeline

The HOPAN study is scheduled to start in January 2023, spanning 2 years.

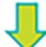

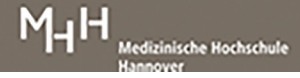

**Fig 2. HOPAN study design.**

## Discussion

### Limitations of the study design

The HOPAN study will use a broad research strategy to identify regional hospice and palliative care networks in Germany. However, since new funding opportunities for hospice and palliative care networks were established in April 2022, it is expected that further networks will emerge during the course of the study. The research team will ask all partners to inform them of all emerging networks, in order for these networks to be invited to participate in the study. Nonetheless, a complete survey of all existing networks cannot be guaranteed.

The network analysis will be based on qualitative and quantitative data collected from network coordinators and leaders. Thus, the HOPAN study will specifically reflect the perspectives of these network coordinators and leaders, while omitting the perspectives of other network members, as well as patients and family caregivers. Hence, the study will not be able to evaluate the impact of hospice and palliative care networks on the quality of patient care.

### Dissemination and implementation

To promote the accessibility and longevity of the research data and results, the research team will report the study findings in a comprehensive and transparent manner. Regardless of the findings, the research team will produce national and international congress presentations and peer-reviewed publications (published as open access, where possible). Data files with no personal identifying information will be kept after the study completion. In accordance with the American Psychological Association Code of Ethics, Sec. 8.14, "Sharing Research Data for Verification" [24], the study leader will not withhold any unidentifiable data from other researchers who wish to verify the conclusions of the author(s). Researchers who wish to use HOPAN study data to answer new research questions must obtain prior permission from the research group and author(s).

The results may be used directly by eligible networks in Germany (of which there are approximately 400) to improve their network structure and work. Based on the level-specific recommendations, each network will receive suggestions for further development and optimisation, tailored to the individual network status, thus enabling a process-oriented network development geared towards regional structures. The specified maturity model will ensure that all relevant network dimensions are considered. The recommendations are expected to support network coordinators' efforts to specify their field of activity, and to be used for training and the further education of network leaders and coordinators in hospice and palliative care. The project results may also be used for quality assessment and quality management in regional hospice and palliative care networks.

## Conclusion

The present study protocol explains the purpose, significance, and scope of the mixed-methods study HOPAN, as well as its study design. The recommendations developed in this study are expected to optimise the establishment and development of regional hospice and palliative care networks in Germany, by ensuring better network coordination and promoting the development of regional hospice and palliative care structures. This improved networking is likely to have a positive effect on the quality of care in hospice and palliative care. The adapted maturity model will be initially published in digital form. If necessary, the model will be further developed in a follow-up project into a digital self-assessment tool for networks, which will automatically show level-specific recommendations relevant to the self-location. In addition, the status analysis and maturity model may be used in the longer term as evaluation tools (e.g. by health insurance providers to fulfil their reporting obligations to the Federal Ministry of Health).

The aim of publishing the present study protocol is to promote transparency by facilitating open access to comprehensive study details that extend beyond the summary publicised in the German Clinical Trials Register. Moreover, the study protocol may serve as a point of reference to the scientific community and other parties interested in the scientific and ethical aspects of the study, and prevent unnecessary duplication.

## Supporting information

**S1 Checklist. STROBE statement—checklist of items that should be included in reports of observational studies–HOPAN study protocol.**
(DOCX)

## Acknowledgments

The authors acknowledge Valerie Appleby's excellent editorial scrutiny of the language of the present study protocol.

## Author Contributions

**Conceptualization:** Sven Schwabe, Christoph Buck, Franziska A. Herbst, Tanja Schleef, Stephanie Stiel, Nils Schneider.

**Funding acquisition:** Sven Schwabe, Christoph Buck.

**Methodology:** Sven Schwabe, Christoph Buck, Franziska A. Herbst, Tanja Schleef, Stephanie Stiel, Nils Schneider.

**Project administration:** Sven Schwabe.

**Supervision:** Nils Schneider.

**Writing – original draft:** Sven Schwabe.

**Writing – review & editing:** Christoph Buck, Franziska A. Herbst, Tanja Schleef, Stephanie Stiel, Nils Schneider.

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
