## [Decision Letter · Decision Letter 0]

10 Apr 2023

PONE-D-22-32383Status exploration and analysis of regional hospice and palliative care networks in Germany: A protocol for a mixed-methods studyPLOS ONE

Dear Dr. Schwabe,

Thank you for submitting your manuscript to PLOS ONE. After careful consideration, we feel that it has merit but does not fully meet PLOS ONE’s publication criteria as it currently stands. Therefore, we invite you to submit a revised version of the manuscript that addresses the points raised during the review process.

ACADEMIC EDITOR: Please find some minor comments from the 2nd reviewer which need to be addressed in the article.No conflicts between both reviews were found.Please submit your revised manuscript by May 12 2023 11:59PM. If you will need more time than this to complete your revisions, please reply to this message or contact the journal office at plosone@plos.org. Please include the following items when submitting your revised manuscript:A rebuttal letter that responds to each point raised by the academic editor and reviewer(s). You should upload this letter as a separate file labeled 'Response to Reviewers'.A marked-up copy of your manuscript that highlights changes made to the original version. You should upload this as a separate file labeled 'Revised Manuscript with Track Changes'.An unmarked version of your revised paper without tracked changes. You should upload this as a separate file labeled 'Manuscript'.If applicable, we recommend that you deposit your laboratory protocols in protocols.io to enhance the reproducibility of your results. Protocols.io assigns your protocol its own identifier (DOI) so that it can be cited independently in the future. For instructions see: https://journals.plos.org/plosone/s/submission-guidelines#loc-laboratory-protocols. Additionally, PLOS ONE offers an option for publishing peer-reviewed Lab Protocol articles, which describe protocols hosted on protocols.io. Read more information on sharing protocols at https://plos.org/protocols?utm_medium=editorial-email&utm_source=authorletters&utm_campaign=protocols.

We look forward to receiving your revised manuscript.

Kind regards,

Stefan Grosek, Ph.D., M.D.,

Academic Editor

PLOS ONE

Journal Requirements:

Additional Editor Comments:

Dear Authors

Please find some minor comments from the 2nd reviewer to be implemnted in your article. Otherwise, this is well written and concise article on status exploration and analysis of regional hospice and palliative care network in Germany.

I'm happy to receive your reply as soon as possible.

Kind regards

Reviewers' comments:

Reviewer's Responses to Questions

**Comments to the Author**

1. Does the manuscript provide a valid rationale for the proposed study, with clearly identified and justified research questions?

Reviewer #1: Yes

Reviewer #2: Yes

2. Is the protocol technically sound and planned in a manner that will lead to a meaningful outcome and allow testing the stated hypotheses?

Reviewer #1: Yes

Reviewer #2: Yes

3. Is the methodology feasible and described in sufficient detail to allow the work to be replicable?

Reviewer #1: Yes

Reviewer #2: Yes

4. Have the authors described where all data underlying the findings will be made available when the study is complete?

Reviewer #1: Yes

Reviewer #2: Yes

5. Is the manuscript presented in an intelligible fashion and written in standard English?

Reviewer #1: Yes

Reviewer #2: Yes

6. Review Comments to the Author

You may also provide optional suggestions and comments to authors that they might find helpful in planning their study.

Reviewer #1: the text is well written. Interesting for the readership. The text is well written. Interesting for the readership. Objective clearly set, well presented, good discussion with references

Reviewer #2: The authors present the protocol of a study examinng hospice - and specialist palliative care networks. The topic is clearly relevant for the establishment of sustainable, efficient and effective collaborations and thus, a state-of-the art cross-sectional care for patients with advanced and life-limiting diseases.

The manuscript is written in a concise, comprehensive and comprehensible manner. Syntax, semantics and wording do not leave room for improvement, but please note, that I am not a native speaker.

The relevant litarature has obviously been researched extensively and choosen meaningfully. A short search performed on my own did not yield any further relevant literature, as far as I can see on first sight.

The methodology is impeccable and checking vs equator network guidelines I could not find major shortcomings.

Yet I have two suggestions:

a) The authors write about "palliative care", also in the title. I believe it should be made clear, that this is rather "specialist palliative care" or "joint general and specialist palliative care". Please consider rewording after discussion in your author group

b) The definition of "hospice" is differing between different international health care settings. You may consider clarifying this not also in the introduction, but even though this is iterative, in the discussion. Please feel free to ignore this suggestion, if your author group (at least the key- authors) agrees to do so after thourough discussion.

7. PLOS authors have the option to publish the peer review history of their article (what does this mean?). If published, this will include your full peer review and any attached files.

Reviewer #1: No

Reviewer #2: No

---

## [Author Response · Author response to Decision Letter 0]

4 May 2023

In the introduction, it was added that the networks include all providers of general and specialised palliative care in the outpatient and inpatient sector.

In the introduction, the term "hospice" in Germany was explained and the hospice care structures were described, referring to the S3- Leitlinie “Palliativmedizin für Patienten mit einer nicht-heilbaren Krebserkrankung”

---

## [Editor Report · Decision Letter 1]

19 May 2023

Status exploration and analysis of regional hospice and palliative care networks in Germany: A protocol for a mixed-methods study

PONE-D-22-32383R1

Dear Dr. Schwabe,

We’re pleased to inform you that your manuscript has been judged scientifically suitable for publication and will be formally accepted for publication once it meets all outstanding technical requirements.

Kind regards,

Stefan Grosek, Ph.D., M.D.,

Academic Editor

PLOS ONE
---

## [Editor Report · Acceptance letter]

24 May 2023

PONE-D-22-32383R1 

Status exploration and analysis of regional hospice and palliative care networks in Germany: A protocol for a mixed-methods study 

Dear Dr. Schwabe:

I'm pleased to inform you that your manuscript has been deemed suitable for publication in PLOS ONE. Congratulations! Your manuscript is now with our production department. 

Kind regards, 

on behalf of

Professor Stefan Grosek 

Academic Editor

PLOS ONE